# Salivary surprise: *Symmerista* caterpillars anoint petioles with red saliva after clipping leaves

David E. Dussourd 🆔 *

Department of Biology, University of Central Arkansas, Conway, Arkansas, United States of America

* dussourd@uca.edu

## Abstract

After feeding on a tree leaf, caterpillars in ten families sever the petiole and allow the remaining leaf fragment to fall to the ground. Previous researchers proposed that the caterpillars thereby reduced bird predation by eliminating visual evidence of feeding. In this study, 26 species of caterpillars in five families were filmed clipping leaves. Caterpillar behavior did not conform to the visual cue hypothesis. Some caterpillars clipped midribs and petioles repeatedly even though a single clip would suffice to reduce visual cues for birds. Every caterpillar that clipped a leaf rubbed its spinneret (which secretes saliva from the labial glands) over the petiole or midrib stub. In the notodontids *Symmerista albifrons* and *S. leucitys*, petiole stubs were bathed in red fluid. Cauterizing the spinneret eliminated fluid application. Dissections documented that the anterior portion of their labial glands contained red pigment, thereby confirming that the red secretion is saliva. When applied to petiole stubs, the red pigment in *Symmerista* saliva travelled several mm in five minutes within the petiole xylem demonstrating the potential for rapid movement of salivary constituents into the plant. In diverse caterpillars, including species that clip leaves, saliva contains substances reported to suppress plant defenses. Thus, leaf clipping likely functions primarily not to remove visual cues, but to introduce salivary constituents into the plant that prevent defenses from being mobilized in nearby leaves where the caterpillar feeds next.

## Introduction

A landmark paper in 1983 by Heinrich and Collins [1] described a novel caterpillar behavior termed leaf or petiole clipping. After partially consuming a leaf, a caterpillar severs the midrib or petiole with its mandibles (Fig 1); the remaining leaf fragment falls to the ground no longer available to the caterpillar. Remarkably, the behavior was found in six families of caterpillars that feed on trees and vines in the eastern U.S. [1–7]. Subsequent studies in Australia and Japan have documented the behavior in an additional four families of caterpillars (Lepidoptera), plus one family of sawflies (Hymenoptera) [8–10]. In the eastern U.S., clipped green leaves can be readily spotted on sidewalks and trails (S1A Fig); the severed petiole documents that the leaf was clipped and not abscised by the plant. During caterpillar outbreaks, the

**Data Availability Statement:** All relevant data are within the manuscript and its Supporting Information files.

**Funding:** This study was supported by the University of Central Arkansas Research Council

(DED). The funders had no role in study design, data collection and analysis, decision to publish, or preparation of the manuscript.

**Competing interests:** The author has declared that no competing interests exist.

**Fig 1. Leaf clipping by tree-feeding caterpillars on excised stems.** (A-C) Final instar *Datana angusii* (Notodontidae: Phalerinae) severing the midrib of a pecan leaflet (*Carya illinoinensis*, Juglandaceae), then applying saliva to the midrib stub (D). (E-G) Final instar *Cecrita biundata* (Notodontidae: Heterocampinae) similarly clipping the petiole of a water oak leaf (*Quercus nigra*, Fagaceae) before secreting saliva onto the petiole stub.

ground may be littered with leaf and midrib fragments (S1B Fig). The quantity of green leaves falling prematurely to the ground, termed greenfall, which includes leaf abscission, has been estimated at 1.3% of total foliar production in a North Carolina forest, a significant loss of photosynthetic material for the trees [6,11]. The proportion of clipped leaves ranged from 25% of these losses in June to over 80% in early October [11].

Why do caterpillars invest resources and take risks to discard potential food? Cutting tough petiole tissues requires time and energy [12]; caterpillar movement and leaf damage provides visual cues, olfactory signals, and vibrations attractive to predators and parasitoids [13–15]. Heinrich and Collins [1] proposed that caterpillars clip leaves to eliminate visual evidence of their feeding. In aviary experiments, black-capped chickadees foraged preferentially on trees with leaves damaged by caterpillars suggesting that birds use feeding damage as a cue in locating prey [1]. Subsequent research has confirmed that leaves with simulated or natural caterpillar damage are attractive not only to avian predators, but also to ants and wasps due to visual and/or olfactory cues [16–18]. Whether petiole stubs are less conspicuous and less attractive to predators than feeding damage has yet to be tested.

Besides eliminating visual cues, leaf clipping might benefit caterpillars by reducing the release of volatile chemicals that attract parasitoids and predators [9,10] or by preventing the induction of defenses in neighboring leaves [6–9]. These hypotheses remain untested, except for a single Australian study on *Perga dorsalis* sawflies [9]. Sawflies blocked from clipping petioles grew faster than controls, an unexpected result if clipping reduces defensive responses. Careful observation of the Australian caterpillars and sawflies revealed that the larvae chewed on petiole stubs for several minutes after clipping a leaf [8], a seemingly unnecessary behavior for discarding damaged leaves. Edwards and Wanjura [8] suggested that perhaps while chewing on the petiole stumps the larvae apply secretions that "neutralize plant defensive substances".

More recently, Dussourd et al. [19] proposed that caterpillars specifically apply saliva to petiole stubs and suggested that petiole clipping serves to expose vascular tissues, thereby allowing salivary constituents and their enzymatic products to enter the plant. Analysis of caterpillar saliva has documented the presence of diverse proteins, phytohormones, and other small

molecules [20–23] that can profoundly alter plant responses including both the production of defensive proteins and allelochemicals [24,25] and the release of volatiles [26–28]. Larvae of *Helicoverpa zea*, for example, apply saliva to leaf edges during feeding [29]. An enzyme in their saliva, glucose oxidase, prevents tobacco plants from increasing levels of nicotine in response to feeding damage [24,30] and causes tobacco stomata to close resulting in decreased emission of volatiles attractive to natural enemies [26].

This paper tests 1) if tree-feeding caterpillars apply saliva to petiole stubs after clipping leaves and 2) if salivary components enter the vascular system. Leaf clipping generally occurs out of sight in the tree canopy. However, caterpillars readily clip leaves in the laboratory when placed on excised stems. My approach was to film caterpillars during and after leaf clipping to determine if saliva is secreted onto petiole stubs. The red saliva of *Symmerista* caterpillars allowed saliva application to be visualized and for the red salivary pigment to be tracked entering the plant.

## Materials and methods

### Study organisms

Thirty-eight species from six major families of tree-feeding macrolepidoptera were studied: swallowtail butterflies (Papilionidae), brush-footed butterflies (Nymphalidae), prominent moths (Notodontidae), owlet moths (Noctuidae), hawk moths (Sphingidae), and giant silk moths (Saturniidae). The caterpillars feed on prominent trees and vines in deciduous forests in the eastern U.S., including oaks, hickories, maples, birches, elms, cherries, etc. [31,32]. Several of the notodontid species, especially *Cecrita guttivitta*, *Lochmaeus manteo*, *Symmerista canicosta*, and *Datana integerrima* are economically important; they undergo periodic outbreaks that can cause extensive defoliation [33–35]. Other species are ecologically significant due to their abundance and widespread distributions [35–37].

Some species were collected as caterpillars in the field in central Arkansas, but most species were acquired by collecting gravid females at mercury vapor and UV lights checked before dawn. Lights were run in central, northwestern, and western Arkansas during summers 2018, 2019, and 2021. In addition, eggs of *Symmerista leucitys* (Notodontidae) were obtained from Marlborough, New Hampshire and some of the *S. albifrons* caterpillars were acquired from Morgantown, West Virginia. Permits were kindly provided by USDA APHIS #P526P-19-00884 for importing *Symmerista* to Arkansas, by the Arkansas Game and Fish Commission (scientific collection permits #051120181, 041820193, and 012120211), DOI National Park Service (permit BUFF-2019-SCI-0014), Arkansas Natural Heritage Commission (permit S-NHCC-21-001), and the Arkansas Department of Parks and Tourism (permits #043–2018 and 050–2019).

### Filming caterpillar behavior

All caterpillars were photographed and filmed during the final instar unless indicated otherwise. Caterpillars were reared on leaves of a tree or vine species reported to be a preferred host [31,32,37–39], then final instars was placed individually on excised stems with ~4–15 leaves. With the stem base in water, the leaves generally remained turgid and green for several days. Caterpillars typically consumed entire leaves except for the midrib, then severed the midrib or petiole before moving to another leaf on the stem. Leaf-clipping was filmed with T4i and T7i Canon cameras on a Gitzo tripod with a Velbon slide rail using a 100mm or MP-E 65mm macro lens with a ring light and panels of LED lights. The number of movies acquired for each species reflected the number of larvae available, the frequency and predictability of leaf clipping, the difficulty of acquiring video of skittish caterpillars, and the amount of time available

for filming. Larvae typically required several hours to as long as a day to finish consuming a single leaf and then clipped the midrib or petiole in just a few minutes. To acquire footage of this brief, but predictable behavior, up to 30 larvae were watched at a time, each on a separate stem; larvae close to finishing a leaf were moved in front of the camera. Disturbing larvae often caused them to remain motionless for lengthy periods, sometimes over an hour. However, caterpillars of many species resumed activity immediately when air was blown over them. Even the slightest breeze that barely moved leaves sufficed to elicit resumed feeding. Predators and parasitoids locate caterpillar prey using visual cues, plant vibrations, and volatiles emitted by the plant and caterpillar frass [4,13–15]. Wind moves and vibrates leaves and disperses volatiles, which should make it harder for enemies to locate prey. These presumed benefits are mostly untested, although Chen et al. [40] found that bird predators, *Parus major*, captured fewer caterpillars under windy conditions.

Host leaves differed greatly in size and in petiole diameter and length. Mature leaves of different species were compared by photographing petioles at 20x, then using image J (National Institutes of Health) to measure petiole width. Leaf area was similarly acquired with image J using scans of entire leaves.

## Leaf clipping by *Symmerista* caterpillars

The three species of *Symmerista* (Notodontidae) in the eastern U.S. are similar in both larval and adult stages. Caterpillars of the maple feeder, *S. leucitys*, were identified by their three black and two white dorsal stripes that extend the full length of the caterpillar [31]. The two oak feeders, *S. albifrons* and *S. canicosta*, are similar to *S. leucitys* at the anterior end, but have five black and four white dorsal stripes at the posterior end. Features of the adult genitalia were used to identify *S. albifrons* [41]. *Symmerista leucitys* larvae were reared on sugar maple, *Acer saccharum* (Sapindaceae), whereas larvae of *S. albifrons* were reared on either white oak, *Quercus alba*, or willow oak, *Quercus phellos* (Fagaceae).

Both *S. leucitys* and *S. albifrons* larvae applied red fluid to petiole stubs. Final instars were dissected to determine if their labial salivary glands were similarly colored. To test if these glands were the only source of red secretion, the spinneret bearing the orifice to the labial glands was cauterized with an ART-E1 electrosurgery unit (Bonart Company, Ltd. New Taipei City, Taiwan) using a sharpened fine tip. A micromanipulator was used to position the cautery tip directly on the spinneret viewed at 50x under a dissecting microscope. After cautery, the *S. albifrons* larvae were filmed to determine if they still applied red secretion when the labial glands could no longer release saliva. None of the four *S. albifrons* larvae that were cauterized produced a silken cocoon documenting that the labial glands (the source of silk) were successfully blocked.

## Do *Symmerista* salivary constituents enter the vascular system?

To test if the red pigment in *Symmerista* saliva moves into the xylem or phloem, *S. leucitys* larvae were allowed to apply saliva after clipping the petiole of a sugar maple leaf (*Acer saccharum*) on an excised stem. Five minutes later, cross sections of the petiole stub were cut by hand with a fresh razor blade and examined for the distinctive red color of saliva. Likewise, cross sections of white oak stubs (*Quercus alba*) were cut 5 minutes after *S. albifrons* larvae clipped the petioles. Petiole sections were examined for red pigment at 50x and photographed at 40x or 100x with an Olympus BX40 compound microscope outfitted with two flashes. Images were assembled with focus stacking using Helicon Focus software version 7.7.5.

To estimate the speed of retrograde (basipetal) flow within clipped petioles, the petioles of five sugar maple leaves on excised stems in the lab were severed 0.5 cm from the leaf blade

using a fresh razor blade, then 1 μl of safranin O dye (0.2%) was placed on the petiole stubs using a Drummond wiretrol applicator (Broomall, Pennsylvania, USA). Safranin O is a small red water-soluble dye (MW = 350.85 g/mol) that has been used previously to study vascular connectivity in trees [42]. After five minutes, the petiole stubs were severed repeatedly from the tip towards the stem at 2.5, 5, 10, 20, 30, 40, and 50 mm with a fresh razor blade. Hand cut slices from these sections were photographed at 100x. During the five-minute interval, the maple stems experienced a photosynthetic photon flux of ~25 μmol/m$^2$/sec as measured with a quantum meter (Apogee Instruments, Model BQM, Logan, Utah, U.S.). White oak stubs were similarly treated with 1 μl safranin O and sliced to determine how far the dye moved in five minutes. The petioles of white oak are much shorter (0.56 ± 0.02 cm) than in sugar maple (7.6 ± 0.5 cm; means ± 1 s.e., N = 25 leaves/species).

The movement of safranin O dye in the xylem could occur through bulk flow or diffusion. To test if diffusion could cause rapid dye movement, five sugar maple leaves were cut from stems and their petiole bases placed in water for 10 minutes. The leaf blades were then removed by cutting the petiole 0.5 cm from the blade. Without leaves, negative pressures generated by transpiration should be reduced in the isolated petioles. A μl of safranin O (0.2%) was added to each petiole stub; five minutes later, the stub was sequentially cut 2.5, 5, 10, 20, and 30 mm from the tip.

Finally, two methods were used to dye vascular tissues to identify which tissues were colored by *Symmerista* saliva and safranin O. First, to dye the xylem, the base of a petiole with leaf still attached was placed in safranin O dye (0.2%) so that the dye was pulled up the xylem by transpiration (one hour for sugar maple, 30 minutes for white oak). Sections of the petiole were then cut just above the dye solution and photographed. Second, to visualize both the xylem and phloem, slices of sugar maple petioles were hand cut and stained with toluidine blue (0.05%) for three minutes followed by safranin O (0.05%) for 30 seconds. Likewise, sections of white oak petioles were stained with astra blue (0.1%) for 12 minutes followed by safranin O (0.05%) for 30 seconds. Safranin O, toluidine blue, and astra blue dye different components of cell walls [43–45]; they are often used to stain plant tissues, including tree petioles [46].

The previous measurements of dye movement in maple and oak petioles were all done indoors using excised stems, which could profoundly affect xylem and phloem transport. To determine if dye moves similarly in sugar maple leaves in the field, the petioles of six leaves on a mature tree were severed 0.5 cm from the leaf blade using a fresh razor blade, then 1 μl of safranin O (0.2%) was placed on the stubs. After 5 minutes, the petiole stubs were cut 1, 10, 20, and 30 mm from the tip and the sections examined for red dye. Sections were then cut further into 1mm slices to more accurately determine the extent of dye movement. The petiole sections were harvested from shaded leaves between 8 and 9:30am during sunny, dry conditions August 23–25, 2021. Photosynthetic photon flux on the leaves before excision averaged 10 ± 3 μmol/m$^2$/sec.

While feeding on sugar maple leaves, *S. leucitys* caterpillars sometimes chewed furrows in leaf veins, applied saliva to the furrows, then continued feeding distal the furrows. To determine if dye applied to furrows moves in the vascular tissues in either the distal or basal direction, artificial furrows were cut in six sugar maple leaves in the field. First, the leaf lamina was removed with scissors on one side of the central vein to simulate feeding, then a notch was cut in the side of the central vein a quarter of the distance from the leaf base to the tip using a fresh razor blade. One μl of safranin O dye (0.2%) was placed on each artificial furrow; five minutes later, the central vein was severed every 10 mm initially and then every 1 mm to determine the distance dye moved. The furrows were cut in shaded leaves on a mature tree in Conway,

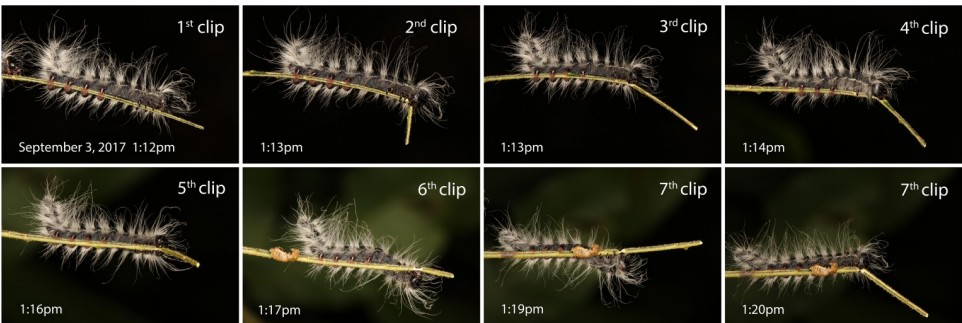

**Fig 2. Final instar of *Datana integerrima* (Notodontidae) clipping a pecan leaflet in the field.** The larva clipped the pecan midrib (*Carya illinoinensis*, Juglandaceae), then backed up and clipped it again repeatedly for a total of seven clips in eight minutes. Each clip was approximately one cm closer to the base of the leaflet.

Arkansas between 8 and 9:30am during dry conditions August 26–29, 2021. Photosynthetic photon flux on the leaves before excision averaged $24 \pm 7$ μmol/m$^2$/sec.

Finally, safranin O dye (0.2%) was also used to estimate the movement of salivary constituents in the midrib of pecan leaflets (*Carya illinoinensis*). Caterpillars of *Datana integerrima* (Notodontidae) consume entire pecan leaflets except for the midrib and then repeatedly clip the midrib. Severing the midrib more than once appears to contradict the hypothesis that clipping serves to introduce salivary components into the plant–unless the components move down the petiole at a faster rate than caterpillars remove sections. If the saliva constituents move more slowly, salivary compounds entering the midrib would fall to the ground when the midrib is clipped again. To estimate movement rates in the field, caterpillar feeding was first simulated with scissors by removing the leaf lamina on both sides of the midrib of six central pecan leaflets, each on a separate leaf. The midribs were then cut at 1/3 the distance to the leaflet tip and 1 μl of safranin O dye (0.2%) was placed on each midrib stub. After 1 minute, sections were removed from the midribs initially at 1, 10, 20, and 30 mm from the tip, then re-cut at 1mm intervals. The dye was allowed only one minute to move into the midrib to match the behavior of a *D. integerrima* larva that was photographed clipping a pecan midrib repeatedly at approximately one-minute intervals (Fig 2). Environmental conditions were similar as when the caterpillar was photographed in the field from 1:12–1:20pm on September 3, 2017 during dry, sunny conditions. Six leaves from the same pecan sapling were used (leaf length $60.3 \pm 2.8$ cm, leaflet length $18.5 \pm 1.3$ cm); dye movement was measured 10:30am - 3pm on August 31, 2021. Photosynthetic photon flux on the leaves averaged $1496 \pm 101$ μmol/m$^2$/sec with temperatures between 31° and 36°C.

## Statistical analyses

Throughout the paper, data are presented as means ± 1 standard error. Statistical analyses were completed with JMP v. 11 using $\alpha < 0.05$ to indicate statistical significance. Data meeting assumptions of normality and equal variance were analyzed with two sample *t*-tests. Otherwise, nonparametric Wilcoxon rank sum tests were used. Categorical data were analyzed with Fisher exact tests due to the small sample sizes.

## Results

### Leaf clipping by *Symmerista* caterpillars

Larvae of *Symmerista leucitys* and *S. albifrons* (Notodontidae) clipped midribs or petioles of their host plants after consuming part or all of a leaf (Fig 3A). The caterpillars applied bright

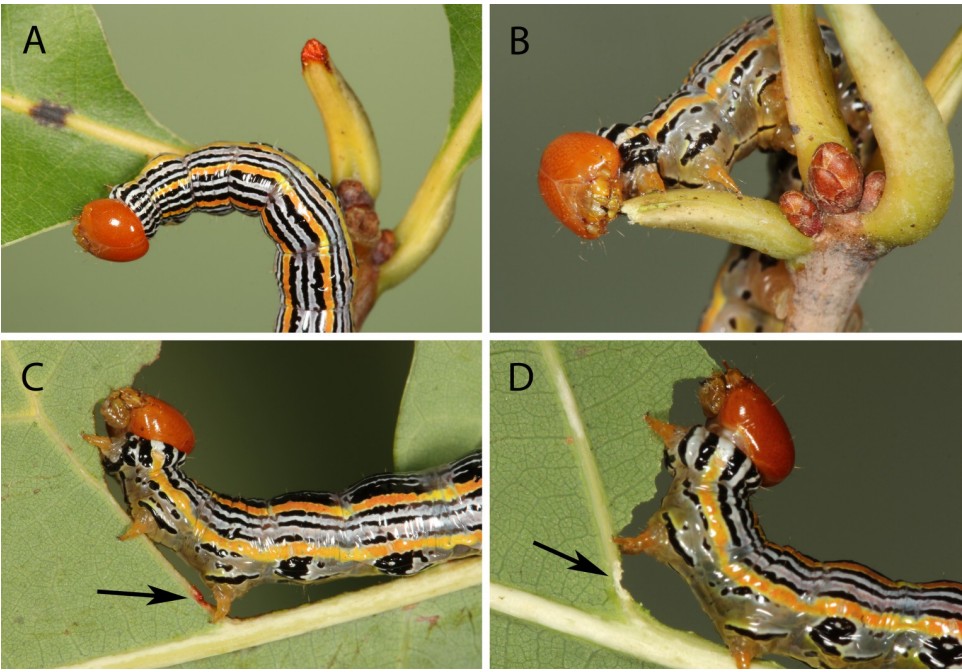

**Fig 3. Intact and cauterized final instar larvae of *Symmerista albifrons* on white oak (*Quercus alba*).** (A) Intact larva with a petiole that it clipped and bathed in red saliva. (B) Larva with a cauterized spinneret that just severed the petiole and is wiping its mouthparts on the stub. No red fluid is visible because cautery blocked the opening of the labial glands preventing release of saliva. Intact (C) and cauterized (D) larvae feeding after chewing a furrow in leaf veins. Red saliva is visible only on the furrow cut by the intact larva.

red fluid, sometimes in copious amounts, to petiole and midrib stubs (S1 Movie). Strands stretching from the spinneret (where the opening of the labial salivary glands is located) to the petiole suggested that the fluid is saliva (S2 Movie). Dissections of final instar *S. leucitys* and *S. albifrons* documented that the labial glands are red in the narrow anterior portion (Fig 4A and 4E); when severed, red pigment exuded from the glands (Fig 4F), which were white when empty. *Symmerista* larvae have paired labial glands that join just before the spinneret, which is located between the two labial palps (Fig 4C). The length of each labial gland approximately equaled or exceeded the length of the caterpillars. The transparent spinneret had a flange-like appearance (Fig 4D), unlike the tubular structures found in many other notodontid caterpillars [22]. Larvae secreted red saliva from the spinneret not only after clipping leaves, but also sometimes with each return stroke during feeding (S5 Movie).

Cauterizing the spinneret of *S. albifrons* eliminated release of red saliva (Fig 3B, S3 Movie) documenting that the red pigment comes only from glands emptying through the spinneret. Four *S. albifrons* larvae produced red secretion before, but not after cautery. Three control larvae that were not cauterized continued to release red fluid until shortly before becoming prepupae ($P = 0.03$ Fisher exact test comparing 0 of 4 cauterized vs. 3 of 3 control larvae that secreted red saliva). Caterpillars with cauterized spinnerets still rubbed their spinnerets on petiole stubs; they even wiped their mouthparts longer than larvae with intact spinnerets (cauterized: n = 2 observations, range 135–229 sec.; intact, n = 11, range 19–104 sec.). The cauterized larvae were not debilitated by the operation; they successfully clipped leaves in a similar amount of time as intact larvae (cauterized: N = 2 observations, range 172–280 sec.; intact: N = 8, range 75–450 sec). The red pigment was absent from the labial glands of prepupae (Fig 4B), which produce the silk used to spin cocoons. Furthermore, cocoon silk was not red

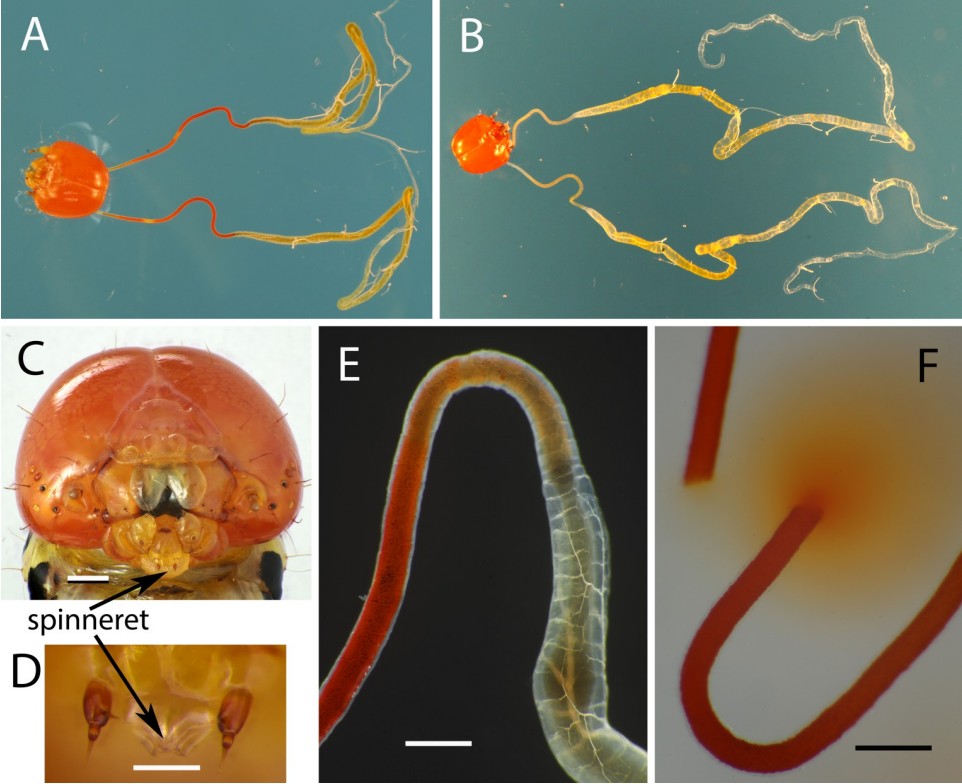

**Fig 4. Labial salivary glands of *Symmerista leucitys*.** (A) Head capsule and attached labial glands of a final instar. (B) Head capsule of a prepupa with paired labial glands that secrete the silk used in constructing a cocoon. (C) Head of a final instar with a close-up of the spinneret (D). (E) Juncture of the thin and thick portions of a labial gland from a final instar. (F) Thin portion of the labial gland from a final instar discharging its red contents after being severed. Scale bars equal 0.5 mm in C, 0.1 mm in D, and 0.3 mm in E and F.

indicating that the red pigment is only produced before larvae stop feeding and prepare for pupation.

## Do *Symmerista* salivary constituents enter the vascular system?

Cross sections of a sugar maple petiole clipped by *S. leucitys* showed salivary pigment in the vascular tissues 2.5 mm from the stub approximately five minutes after clipping (Fig 5A). Four other petioles and midribs clipped by *S. leucitys* were similarly stained with red pigment at least 1mm and as much as 4 mm from the stub 5 minutes after larvae finished applying saliva. Likewise, seven white oak petioles clipped by *S. albifrons* showed pigment 4 ± 1 mm from the stub 5 minutes after clipping (S2A Fig). Salivary components clearly can move quickly into the vascular system during or after leaf clipping. To determine if the xylem or phloem was stained, petioles of excised sugar maple leaves were placed in safranin O dye for an hour; transpiration from the leaves pulled the dye into the xylem matching the location of *Symmerista* pigment (Fig 5B). When 1 µl safranin O dye was instead placed on sugar maple petiole stubs, the dye moved down the xylem over 1 cm in five minutes in all five leaves tested and as far as 6 cm (Fig 5C). The safranin O dye did not travel as far in isolated petioles with no adjacent leaves and thus reduced transpiration; the dye moved just 2.5 to 5 mm in five minutes documenting that diffusion is inadequate to account for rapid movement. The distance that the dye travelled in petioles with adjacent leaves was significantly greater than in isolated petioles ($P = 0.0088$,

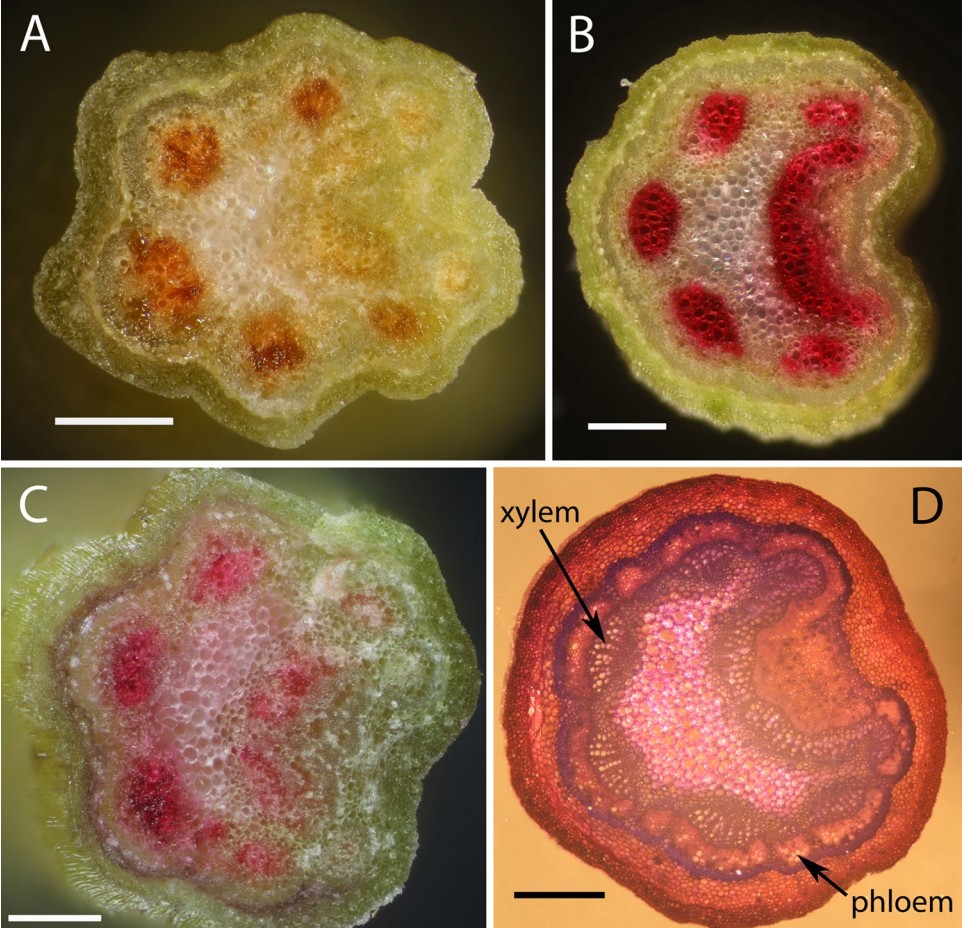

**Fig 5. Petioles of sugar maple (*Acer saccharum*) with xylem stained by *Symmerista leucitys* saliva or dye.** (A) Petiole clipped by a *S. leucitys* larva, then severed with a razor blade 2.5 mm from the stub tip approximately five minutes after the larva finished applying saliva. The red salivary pigment moved into the petiole staining the xylem. (B) Petiole xylem stained by safranin O dye (0.2%) pulled up the xylem by transpiration. (C) Petiole stub treated with 1μl safranin O dye (0.2%). After 5 minutes, the petiole was severed 2.5mm from the initial cut. The dye moved in a retrograde direction down the xylem staining the same tissues as *Symmerista* pigment in (A) and dye in (B). The petiole slices in A-C were photographed dry to prevent pigment and dye from dissolving or spreading to adjacent tissues. (D) Thin slice of petiole stained with toluidine blue followed by safranin O to show the location of xylem and phloem. Scale bars equal 0.2 mm in A-C and 0.5mm in D. The dorsal surface of the petioles is on the right side.

Wilcoxon rank sum test). The retrograde flow of dye in the xylem was not an artifact of testing excised stems indoors. Safranin dye applied to severed sugar maple petioles in the field also moved a substantial distance (1.9 ± 0.1 cm; N = 6 petioles) over a 5 minute period, even under low light conditions in early morning (S3 Fig).

## Saliva application in other leaf-clipping caterpillars

Of the 38 caterpillar species in this study (including *Symmerista*), 34 species in six families clipped the petiole or midrib after partially consuming a leaf (Fig 1, Table 1). For 30 of the 34 species, leaf clipping had not been previously reported (Table 1 species #1–10, 14–27, 29–34). After the caterpillars finished feeding, they backed down the midrib, then used their mandibles to eat through the midrib or petiole (often at their juncture) causing the distal portion to fall to the ground with the caterpillar still attached to the plant. In the lab, caterpillars typically

**Table 1. Time that caterpillars in 34 species spent clipping leaves and applying saliva.**

| Caterpillar species # | Caterpillar family (blue) and subfamily[a] | Caterpillar species[a] | Host plant species | Host plant family | Time clipping leaf (sec.)[b] | Time applying saliva (sec.)[b] | Movie # |
|---|---|---|---|---|---|---|---|
| | **Papilionidae** | | | | | | |
| 1 | Papilioninae | *Papilio troilus* | *Sassafras albidum* | Lauraceae | - | - | |
| | **Nymphalidae** | | | | | | |
| 2 | Limenitidinae | *Limenitis arthemis* | *Prunus serotina* | Rosaceae | 170 (2) | 127 ± 51 (5) | S8 |
| 3 | Apaturinae | *Asterocampa celtis* | *Celtis laevigata* | Cannabaceae | - | - | |
| 4 | Apaturinae | *Asterocampa clyton* | *Celtis laevigata* | Cannabaceae | - | - | |
| | **Notodontidae** | | | | | | |
| 5 | Notodontinae | *Paraeschra georgica* | *Quercus nigra* | Fagaceae | 65 (1) | 45 (2) | S9 |
| 6 | Cerurinae | *Furcula borealis* | *Prunus serotina* | Rosaceae | 104 ± 24 (4) | 99 ± 7 (7) | S10 |
| 7 | Phalerinae | *Datana angusii* | *Carya illinoinensis* | Juglandaceae | 58 ± 9 (14) | 19 ± 4 (17) | S11 |
| 8 | Phalerinae | *Datana integerrima* | *Carya illinoinensis* | Juglandaceae | 24 ± 3 (10) | 3 ± 1 (15) | |
| 9 | Phalerinae | *Datana perspicua*[c] | *Rhus glabra* | Anacardiaceae | - | - | |
| 10 | Phalerinae | *Peridea angulosa* | *Quercus nigra* | Fagaceae | 76 ± 7 (5) | 25 ± 12 (5) | |
| 11 | Heterocampinae | *Cecrita biundata* | *Quercus nigra* | Fagaceae | 145 ± 24 (10) | 79 ± 10 (13) | |
| 12 | Heterocampinae | *Cecrita guttivitta* | *Quercus nigra* | Fagaceae | 82 ± 15 (8) | 52 ± 5 (14) | |
| 13 | Heterocampinae | *Coelodasys unicornis* | *Betula nigra* | Betulaceae | 351 ± 19 (4) | 137 ± 19 (6) | S12 |
| 14 | Heterocampinae | *Heterocampa obliqua* | *Quercus nigra* | Fagaceae | 41 ± 5 (3) | 6 ± 2 (3) | |
| 15 | Heterocampinae | *Heterocampa umbrata*[d] | *Quercus nigra* | Fagaceae | 302 ± 179 (3) | 43 ± 15 (5) | |
| 15 | Heterocampinae | *Heterocampa umbrata* | *Quercus nigra* | Fagaceae | 74 (1) | 30 ± 5 (6) | |
| 16 | Heterocampinae | *Lochmaeus bilineata* | *Ulmus americana* | Ulmaceae | 623 ± 68 (4) | 84 ± 23 (5) | |
| 16 | Heterocampinae | *Lochmaeus bilineata* | *Ulmus crassifolia* | Ulmaceae | 194 ± 18 (3) | 64 ± 17 (3) | |
| 17 | Heterocampinae | *Lochmaeus manteo* | *Quercus nigra* | Fagaceae | 105 (2) | 17 ± 4 (5) | |
| 18 | Heterocampinae | *Macrurocampa marthesia* | *Quercus shumardii* | Fagaceae | 155 (2) | 65 ± 11 (5) | |
| 19 | Heterocampinae | *Misogada unicolor* | *Platanus occidentalis* | Platanaceae | - | - | |
| 20 | Heterocampinae | *Oedemasia leptinoides* | *Carya illinoinensis* | Juglandaceae | 121 (2) | 42 ± 14 (4) | |
| 21 | Heterocampinae | *Schizura ipomoeae* | *Acer rubrum* | Sapindaceae | 148 (1) | 68 ± 12 (4) | |
| 22 | Nystaleinae | *Symmerista albifrons* | *Quercus alba* | Fagaceae | 249 ± 42 (8) | 48 ± 8 (11) | S1, S2 |
| 23 | Nystaleinae | *Symmerista leucitys* | *Acer saccharum* | Sapindaceae | 138 ± 25 (6) | 46 ± 9 (6) | |
| | **Noctuidae** | | | | | | |
| 24 | Acronictinae | *Acronicta hasta* | *Prunus serotina* | Rosaceae | - | - | |
| 25 | Acronictinae | *Acronicta impleta* | *Carya illinoinensis* | Juglandaceae | 427 (1) | 190 (2) | S13 |
| 26 | Acronictinae | *Acronicta morula* | *Ulmus americana* | Ulmaceae | - | - | |
| | **Sphingidae** | | | | | | |
| 27 | Sphinginae | *Ceratomia amyntor* | *Ulmus americana* | Ulmaceae | 78 (2) | 96 ± 22 (4) | S14 |
| 28 | Sphinginae | *Ceratomia undulosa* | *Fraxinus pennsylvanica* | Oleaceae | 63 (2) | 35 ± 7 (7) | |
| 29 | Sphinginae | *Dolba hyloeus* | *Ilex decidua* | Aquifoliaceae | 134 (1) | 43 (1) | |
| 30 | Smerinthinae | *Amorpha juglandis* | *Carya illinoinensis* | Juglandaceae | - | - | |
| | **Saturniidae** | | | | | | |
| 31 | Ceratocampinae | *Anisota virginiensis* | *Quercus shumardii* | Fagaceae | 182 (1) | 85 (1) | S15 |
| 32 | Hemileucinae | *Automeris io* | *Prunus serotina* | Rosaceae | 79 ± 9 (6) | 76 ± 14 (6) | S16 |
| 33 | Saturniinae | *Actias luna*[d] | *Carya illinoinensis* | Juglandaceae | 106 (2) | 53 ± 11 (3) | S17 |

*(Continued)*

**Table 1.** (Continued)

| Caterpillar species # | Caterpillar family (blue) and subfamily[a] | Caterpillar species[a] | Host plant species | Host plant family | Time clipping leaf (sec.)[b] | Time applying saliva (sec.)[b] | Movie # |
|---|---|---|---|---|---|---|---|
| 34 | Saturniinae | *Hyalophora cecropia* [d] | *Prunus serotina* | Rosaceae | 214 ± 122 (4) | 90 ± 8 (4) | S18 |

Time clipping leaf includes only the amount of time that larvae spent cutting through a petiole, midrib or stem once, plus any time devoted to chewing on the stub. It does not include time larvae spent selecting a location to clip, which was sometimes a lengthy procedure. After the petiole, midrib or stem was severed, the larvae applied saliva. Larvae often started to depart, then returned to secrete more saliva. Only the time rubbing the spinneret on the stub was included, not time moving to and from the stub or time applying saliva while clipping the petiole/midrib. Filming often began after larvae initiated clipping and thus more data are available for time spent applying saliva than time clipping the petiole or midrib. All data are for final instars unless indicated otherwise.

[a] Caterpillar species are classified according to recent studies [47–50].

[b] means ± s.e., number of replicates (movie clips) is shown in parentheses.

[c] 4th of 6 instars; larvae were observed on intact plants in the field.

[d] penultimate instar.

consumed entire leaf blades except for part of the midrib. In nature, larvae often eat only a portion of the leaf blade before severing the petiole (S1A Fig).

Video of leaf clipping was acquired with 26 species (including *Symmerista*); every single larva in the 26 species rubbed their spinneret over the midrib or petiole stub after clipping a leaf. Representative movies are provided in supplemental resources and listed in Table 1. The movies document the striking similarity in the behavior of caterpillars across 11 subfamilies within five families. As with *Symmerista*, saliva release was often visible during and especially after leaf clipping; strands of saliva could at times be seen stretching from the petiole to the caterpillar's spinneret (S7 Movie). The average time spent cutting through petioles or midribs in different species varied from 24 to 623 seconds; the average time spent applying saliva ranged from 3 seconds for *Datana integerrima* to 137 seconds for *Coelodasys unicornis* (Table 1). As described below, *D. integerrima* clipped midribs repeatedly, rubbing their spinneret only briefly over midrib stubs after each clip. In contrast, *C. unicornis* larvae on river birch, *Betula nigra*, clipped petioles once and meticulously painted the petiole stubs with saliva over and over again (S12 Movie). Averaging across all 26 species, caterpillars spent 154 ± 25 seconds clipping midribs or petioles and 64 ± 8 seconds applying saliva. Several species were not filmed successfully, particularly species that fed primarily at night. However, leaf fragments with severed petioles provided evidence of their leaf clipping (Table 1 species #1, 3, 4, 19, 24, 26, 30).

Besides *Symmerista*, only one other species in this study, *C. unicornis*, sometimes produced colored petiole stubs. The salivary glands of dissected final instars were not brightly colored suggesting that the rusty coloration of petiole stubs is probably due to an enzymatic (browning) reaction and not to salivary pigments.

To film leaf clipping, it was sometimes necessary to remove leaves that blocked the view. Interestingly, caterpillars of *Paraeschra georgica* (Notodontidae) on water oak, *Quercus nigra*, often found the wound where the petiole had been attached; the larvae rubbed their spinneret on the wound as though they had clipped the leaf. In other species, including *Heterocampa biundata* (Notodontidae), larvae sometimes revisited previous petiole, midrib, or stem clips to wipe their spinneret over the stubs again.

## Variation in leaf-clipping behaviors

In some species, caterpillars clipped all or nearly all leaves that they fed upon (Table 1 species #1, 2, 6–8, 10–13, 15, 16, 22). In other species, only some of the leaves were clipped (Table 1

species #3, 4, 9, 17, 18, 20, 21 on *Acer rubrum*, 24–28, 30–34). Larvae of *Macrurocampa marthesia* (Notodontidae), for example, clipped 19 of 37 leaves that they fed on. Finally, other species clipped leaves infrequently (Table 1 species #5, 14, 19, 23, 29). For example, ten *Symmerista leucitys* larvae fed on 40 sugar maple leaves over a three-day period, but only clipped two of them. Instead of clipping, the *S. leucitys* larvae often consumed a section of a leaf, rubbed saliva on the main vein in the section, then resumed feeding elsewhere on the leaf (S1D Fig). Seven *S. albifrons* larvae on white oak, in contrast, clipped almost all leaves that they fed on over a three-day period (18 of 23 leaves). On smaller white oak leaves with less robust veins, the *S. albifrons* larvae sometimes consumed the entire leaf down to the petiole, then applied saliva on the petiole stub without clipping. Several other species, including various notodontids, noctuids, sphingids, and saturniids, similarly ate entire leaves including the midrib, then rubbed saliva on the remaining petiole stub (Table 2, S19 Movie). Thus, even caterpillars that did not clip leaves still applied saliva to midrib or petiole stubs before leaving a leaf (although sometimes saliva application was brief, especially with *Ceratomia undulosa*). On average, the 11 species with non-clipping caterpillars spent 31 ± 6 seconds applying saliva to petiole or midrib stubs (Table 2). Stems with non-clipping caterpillars often resembled stems with clipped leaves; in both cases, only petiole stubs remained (S1C Fig).

Remarkably, caterpillars of some species clipped a midrib or petiole more than once. Both *Datana integerrima* and *D. angusii* larvae often clipped the midrib of a pecan leaflet repeatedly in the lab. In the field, individual *D. integerrima* larvae were observed clipping a pecan midrib up to nine times, rubbing their mouthparts over the midrib stub each time (Fig 2). Each clipped section was approximately 1 cm long. To determine if *Datana* salivary constituents might move into the xylem of pecan as documented previously with maple and oak leaves, six pecan leaflets were trimmed on both sides of the midrib to simulate feeding by *D. integerrima*. The midribs were then clipped and safranin O dye applied to the midrib stubs. On average, the dye moved 2.3 ± 0.3 cm in 1 minute, approximately the time between successive vein clips by

**Table 2. Time that non-clipping caterpillars spent applying saliva to petiole or midrib stubs.**

| Caterpillar family (blue) and subfamily | Caterpillar species | Host plant species | Host plant family | Time applying saliva (sec.) [a] |
|---|---|---|---|---|
| **Notodontidae** | | | | |
| Notodontinae | *Paraeschra georgica* | *Quercus nigra* | Fagaceae | 18 ± 5 (8) |
| Heterocampinae | *Cecrita biundata* | *Quercus nigra* | Fagaceae | 35 (2) |
| Heterocampinae | *Heterocampa obliqua* | *Quercus nigra* | Fagaceae | 19 ± 9 (10) |
| Heterocampinae | *Oedemasia leptinoides* | *Carya illinoinensis* | Juglandaceae | 7 (1) |
| Heterocampinae | *Schizura ipomoeae* | *Quercus phellos* | Fagaceae | 24 ± 15 (3) |
| **Noctuidae** | | | | |
| Acronictinae | *Acronicta hasta* | *Prunus serotina* | Rosaceae | - |
| **Sphingidae** | | | | |
| Sphinginae | *Ceratomia undulosa* | *Fraxinus pennsylvanica* | Oleaceae | 27 ± 11 (3) |
| Sphinginae | *Dolba hyloeus* | *Ilex decidua* | Aquifoliaceae | 56 ± 7 (11) |
| Macroglossinae | *Darapsa myron* | *Ampelopsis cordata* | Vitaceae | 11 ± 5 (7) |
| **Saturniidae** | | | | |
| Hemileucinae | *Automeris io* [b] | *Celtis laevigata* | Cannabaceae | 72 ± 18 (3) |
| Saturniinae | *Actias luna* [b] | *Carya illinoinensis* | Juglandaceae | 33 (2) |
| Saturniinae | *Hyalophora cecropia* [b] | *Prunus serotina* | Rosaceae | 41 (2) |

All data are for final instar caterpillars unless indicated otherwise.

[a] means ± s.e., number of replicates (movie clips) is shown in parentheses.

[b] penultimate instar.

the *D. integerrima* larva photographed in Fig 2. Thus, salivary components could potentially move in a retrograde direction faster down the petiole than *Datana* larvae cut segments.

Both *D. integerrima* and *D. angusii* are social species that feed gregariously (S4 Fig). The ground under *Datana* aggregations was littered with dozens or hundreds of pieces of midribs and side veins. Another social species, *Automeris io* (Saturniidae) on black cherry and hackberry, also sometimes clipped midribs, petioles, and stems repeatedly even when larvae were isolated on cuttings in the lab (S16 Movie). Several solitary species (Table 1 species #2, 6, 16, 28) also sometimes clipped a leaf twice. Caterpillars that clipped leaves more than once usually spent more time cutting and more time applying saliva with basal clips. In an extreme example, two *Liminitis arthemis* larvae (Nymphalidae) each spent just 3 seconds applying saliva after clipping the midrib, then 186 and 200 seconds applying saliva after clipping the petiole of the same leaves.

Only a few caterpillar species were filmed in the lab on multiple host species. Their behavior often differed on different hosts. *Lochmaeus bilineata* larvae, for example, took significantly longer to clip large American elm leaves (*Ulmus americana*, 623 ± 68 seconds) than much smaller cedar elm leaves (*Ulmus crassifolia*, 194 ± 18 seconds) (*P* = 0.0033, two-sample *t*-test). Petiole diameter, petiole length, and leaf area were all significantly greater in American elm (*P* < 0.0001, Wilcoxon rank sum tests, S3 Data). The behavior of *Schizura ipomoeae* also varied. On red maple (*Acer rubrum*), 7 of 11 larvae clipped petioles, whereas on willow oak (*Quercus phellos*), 0 of 7 clipped, a significant difference (*P* = 0.013, Fisher exact test). Red maple leaves have petioles that are substantially longer (74 ± 4 mm) and wider (1.35 ± 0.04 mm) than willow oak petioles (length 3.80 ± 0.13 mm, diameter 1.05 ± 0.02 mm, n = 25 leaves/species) (*P* < 0.0001 Wilcoxon rank sum tests). On willow oak, the larvae consumed entire leaves including the narrow midribs, then wiped their spinneret on the remaining petiole stubs.

Whereas most species appeared to clip leaves with minimal difficulty, slippery round midribs and petioles were sometimes difficult for larvae to grip and penetrate. One final instar of *S. albifrons*, for example, was unable to cut into the midrib of a white oak leaf (S6 Movie) despite four attempts at one location totaling over 8.8 minutes. For some caterpillars, the time and energy required for clipping may be costly.

Not all larvae consumed entire leaves or clipped petioles. Two nocturnal species, *Catocala maestosa* on *Carya illinoinensis* and *Zale lunata* on *Prunus serotina* and on *Salix*, tattered leaves with their feeding, then rested during the day in nearby hideaways provided for them. Neither of these two members of the Erebidae (Erebinae) was observed to clip leaves on excised stems in the lab. Clipping may not be necessary for such highly mobile species that spend the day away from their leaf damage and can easily move to new feeding sites at night. Another member of the Erebidae, *Dasychira tephra* (Lymantriinae), on three oak species (*Quercus shumardii*, *Q. nigra*, *Q. phellos*) also was not observed clipping leaves.

## Other caterpillar behaviors

While feeding on a leaf, caterpillars often chewed one or more furrows in a midrib or side vein (Fig 6A and 6C; Table 1 species #7, 8, 11, 12, 15, 20, 22, 23). The larvae then continued feeding beyond the furrow (S20 Movie), unlike in leaf clipping where the remaining leaf fragment drops from the plant. Immediately after cutting a furrow, caterpillars invariably rubbed their spinneret over the furrow. Final instars of *Heterocampa umbrata*, for example, spent an average of 31 seconds wiping their spinneret (n = 2 larvae filmed). *Symmerista leucitys* and *S. albifrons* larvae both painted furrows red documenting that saliva is applied to furrows (Fig 3C). The saliva was secreted primarily onto the basal side of the furrow (S4 Movie). Cauterizing

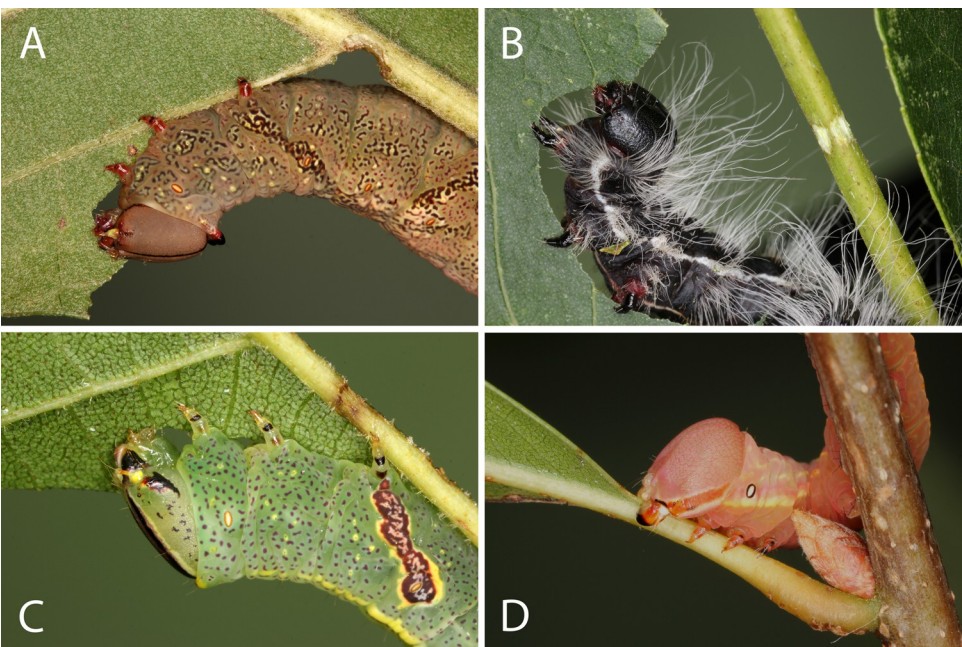

**Fig 6. Furrows, girdles, and petiole crimps of final instar notodontids.** (A) *Heterocampa umbrata* larva feeding on southern red oak (*Quercus falcata*) after chewing a furrow in the midrib. (B) *Datana integerrima* larva feeding on a pecan leaflet (*Carya illinoinensis*) near a girdle that it cut in the rachis. (C) *Cecrita guttivitta* larva feeding near a furrow in a *Carya illinoinensis* leaf. (D) *Peridea angulosa* crimping the petiole of a water oak leaf (*Quercus nigra*).

spinnerets of *S. albifrons* eliminated release of red fluid onto furrows, just like with leaf clipping (Fig 3D). When safranin O (0.2%) was applied to simulated furrows cut in sugar maple midribs with a razor blade in the field, the red dye moved in both basal (11 ± 2 mm) and distal (12 ± 3 mm) directions.

Other caterpillar species chewed girdles, which were grooves that extended around a stem, rachis, or petiole (Fig 6B, Table 1 species #7, 8, 13, 20, 21). Girdling has been studied primarily in a notodontid *Oedemasia leptinoides* (formerly *Schizura*); the larvae cut girdles around pecan rachises, then bathe the girdle surface with saliva [12,19,51]. Other girdlers in this study similarly secreted saliva onto girdles (S21 Movie). *Schizura ipomeae* filmed on river birch, *Betula nigra*, spent an average of 990 seconds cutting girdles (n = 2 girdles) and 46 ± 18 seconds applying saliva to girdles (n = 4 girdles). To simulate girdles, a wirestripper was used to remove peripheral tissues in a ring cut in river birch stems and in pecan rachises. Safranin O dye applied to these artificial girdles did not move in either a basal or distal direction unless the xylem was cut by the wirestripper. Girdles produced by *O. leptinoides* in river birch and pecan do not appear to sever the main xylem vessels [51] suggesting that only salivary components able to cross cell walls would enter the xylem. However, pecan rachises have an additional small auxillary vascular bundle that *O. leptinoides* larvae cut; it provides a potential point of entry for salivary constituents into the xylem in this species.

Finally, some caterpillar species used their mandibles to crimp petioles or midribs repeatedly (Fig 6D, Table 1 species #5, 6, 9, 10, 17, 21, 24, 26) [52]. The larvae did not cut into the plant surface as in girdling. In *Peridea angulosa*, larvae crimped the petioles of up to 10 leaves in succession before initiating feeding, apparently preparing the leaves for future consumption. *Oedemasia leptinoides* larvae likewise cut multiple girdles and then feed distal to the girdles over a period of days [12, 51, DE Dussourd Unpublished]. Caterpillars pinched petioles from

different angles, sometimes in a ring that resembled a girdle (S22 Movie). In *Schizura ipomoeae*, crimping was followed by brief saliva application (S22 Movie). The behavior of different species appeared to vary across a continuum from crimping without saliva application, to crimping in a ring that is daubed with saliva, to cutting and pinching girdles that are bathed in copious quantities of saliva. Many of the caterpillars filmed in this study have not been previously reported to crimp petioles (species #6, 21, 24, 26 in Table 1), cut girdles (species #7, 8), or chew furrows (species #7, 8, 11, 22, 23).

## Discussion

### Distribution of leaf-clipping behaviors

The observations in this study support previous reports by Heinrich and Collins [1] and others that leaf clipping has a broad distribution. Caterpillars in five families (Nymphalidae, Notodontidae, Noctuidae, Sphingidae, Saturniidae) were filmed clipping petioles or midribs; larvae from one additional family (Papilionidae) left evidence of clipped leaves under their feeding. These families are classified in three superfamilies (Papilionoidea, Noctuoidea, Bombycoidea) in current phylogenies [49] documenting the widespread distribution of the behavior. Observations by others [8,10] add four more families of leaf clippers (Limacodidae, Geometridae, Lasiocampidae, Anthelidae) representing three additional superfamilies (Cossoidea, Geometroidea, Lasiocampoidea). The most recent common ancestor of the seven leaf-clipping superfamilies dates back over 110 million years to the Cretaceous [49]. The presence of clipping also in sawflies (Hymenoptera) in Australia suggests that the presence of leaf clipping in distantly related groups likely represents not a single early origin but convergent evolution of the same behavior in multiple groups. However, the widespread distribution of leaf clipping in the Notodontidae, Sphingidae, and Saturniidae suggests that the trait likely evolved early in the evolution of each of these families. So far, leaf clipping has only been documented in the eastern U.S., Australia, and Japan [1,6–10,51]; comparable studies elsewhere in the world, particularly in the tropics, may greatly expand the list of clipping species. The main leaf-clipping groups (Notodontidae, Sphingidae, and Saturniidae) all have high diversity in tropical forests [53,54].

Only four of the 36 caterpillar species in this study were not observed clipping leaves (the sphingid *Darapsa myron* and all three members of the Erebidae). Leaf clipping by Erebidae has been reported by others, including in species of *Catocala* and *Zale* [1,2,6]. Given that leaf clipping is facultative in many caterpillar species (Table 2) [1,8], the absence of clipping under laboratory conditions does not preclude the possibility that *D. myron* and the erebids in this study sometimes clip leaves in nature.

Plants that caterpillars clipped are classified in 12 families in eight orders that are broadly distributed among angiosperm groups, including members of the magnoliids, fabids, malvids, campanulids, and lamiids according to the classification of the Angiosperm Phylogeny Group [55]. That distantly related caterpillars show such remarkably similar behavior on diverse plants suggests a functional similarity in their hosts. Curiously, leaf clipping has not been reported for caterpillars on herbaceous plants, only for larvae feeding on trees and vines [1,8,51]. Heinrich [5] suggested that discarding leaf remnants, seemingly wasteful feeding, would be less advantageous on herbs with fewer leaves available. Leaf clipping might also be unnecessary on forbs; their higher water content [56] could allow liberal release of saliva throughout feeding. Forb leaves also are less tough than leaves on woody plants [57], which might permit forb-feeders to eat entire leaves including the midrib. Whether larvae on forbs apply saliva to petiole stubs after consuming whole leaves, as documented with several tree-feeding species in this study (Table 2), has not been reported.

## Function of leaf-clipping: Visual cue hypothesis

Heinrich and Collins [1] proposed that caterpillars clip petioles to reduce visual evidence of feeding. Caterpillars that cut off petioles at the base, such as many *Ceratomia amyntor* (Sphingidae) (S14 Movie), do eliminate all visual evidence of the leaf and petiole, consistent with the visual cue hypothesis. But in other species, petiole stubs provide seemingly conspicuous evidence of the caterpillar's presence.

Four lines of evidence presented in this paper are not consistent with the visual cue hypothesis. To reduce evidence of feeding, caterpillars only need to clip leaves once at the base, but *Datana* larvae clipped midribs repeatedly (Fig 2). This social species could conceivably clip midribs in sections to reduce the chance of dropping siblings to the ground, a plausible outcome if a lengthy midrib was cut only once at the base. However, solitary larvae of other species (*Lochmaeus bilineata*, *Ceratomia undulosa*) and isolated larvae of social species (*Datana* spp. and *Automeris io*) also clipped petioles/midribs more than once suggesting that clipping leaves repeatedly has an alternative, as yet unknown, function.

Second, leaf-clipping caterpillars invariably rubbed their spinneret over petiole stubs, an unnecessary behavior if the goal is to eliminate visual evidence of feeding. Caterpillars that consumed entire leaves without clipping the petiole also applied saliva while rubbing their spinneret on the petiole tip (Table 2, S19 Movie). Several of the species that clipped leaves also chewed furrows in midribs and/or cut girdles. Neither furrows nor girdles reduced visual signs of caterpillar feeding, but saliva was applied to these cuts as well suggesting a common function of leaf clipping, furrowing, and girdling.

Third, caterpillars that consumed whole leaves including the midrib sometimes clipped off a short section of the petiole (S12 Movie). Shortening the petiole had minimal effect on the visibility of leaf damage since the entire blade was already gone.

Finally, the suggestion [1,3,4] that spiny and/or aposematic caterpillar species do not clip leaves was not supported. If leaf clipping serves to help caterpillars hide, then well-defended caterpillars with bright coloration would presumably not need to eliminate evidence of feeding. But *Symmerista*, *Datana*, and *Anisota* all clipped leaves despite their aposematic coloration and occurrence in groups that are highly conspicuous (S4 Fig, S1, S11 and S15 Movies). Likewise, larvae of *Automeris io* with stinging spines and *Hyalophora cecropia* with colorful spiny knobs both clipped leaves (S16 and S18 Movies). Edwards and Wanjura [8] also reported that leaf-cutting species are not all inconspicuous. The larvae of the leaf-clipping limacodid *Doratifera quadriguttata*, for example, have bright coloration and stinging hairs.

## Function of leaf clipping: Saliva hypothesis

The discovery that *Symmerista* caterpillars produce red saliva and apply it to petiole stubs after clipping leaves strongly supports the alternative saliva hypothesis. Dissections confirmed that the red pigment occurs within the labial glands. When the opening of the labial glands was cauterized, *S. albifrons* larvae no longer released red saliva demonstrating that secretions from the spinneret are the only source of red fluid. The function and identity of the red pigment in saliva is not known.

Other caterpillar species did not have colored saliva, but they showed the same spinneret-rubbing behavior as *Symmerista*. Every caterpillar that was filmed clipping a leaf wiped their mouthparts over the petiole or midrib stub. In some movies, the surface of the petiole or midrib became reflective after the caterpillar's mouthparts passed over it documenting that fluid was applied. Strands extending directly from the spinneret to the leaf surface demonstrated that the secretion was saliva (S7 Movie). *Symmerista* caterpillars also applied red saliva to furrows cut in leaf veins. Again, cauterizing the spinneret eliminated secretion of red fluid. Other

notodontid caterpillars filmed girdling (*Coelodasys unicornis* and *Schizura ipomoeae*) similarly wiped their spinneret on girdles and applied fluid (S21 Movie). Previous studies with another notodontid, *Oedemasia leptinoides*, documented that this fluid is saliva [19]. Thus, leaf clips, furrows and girdles may all share a common function—introducing salivary components and their enzymatic products into freshly exposed vascular tissues. Australian caterpillars and saw-flies that were observed "chewing" on petioles after clipping a leaf [8] may likewise secrete saliva while rubbing their mouthparts over petiole stubs.

The cues that trigger leaf clipping, furrowing and girdling in caterpillars have not been identified. However, when I removed leaves for filming, *Paraeschra georgica* caterpillars often found the wounds where the petioles had been attached and rubbed saliva on them. Saliva application after leaf clipping is clearly not just a programmed sequence of behaviors. The caterpillars detect some cue, presumably volatiles, that triggers saliva secretion. Likewise, the observation that *S. albifrons* larvae with cauterized spinnerets rubbed their mouthparts over petiole stubs longer than normal suggests that they can detect saliva constituents or changes induced by saliva application to decide when to stop applying saliva.

## Effect of saliva on tree physiology

When caterpillars clip leaves or chew furrows and girdles, they cause seemingly unnecessary damage. Plants respond to wounds with complex biochemical and physiological changes that can include chemical and electrical signals, up and down regulation of genes, increased production of defensive enzymes and metabolites, and enhanced release of volatiles attractive to natural enemies [15,58–60]. Damage to the midrib and side veins, especially near the leaf base, elicits particularly robust increases in defensive responses [61–64] and decreases in photosynthesis [65–68]. Cuts in aspen leaves *Populus tremula*, for example, caused a burst in emission of volatiles that was six times higher when the midrib was cut than when only the lamina was severed [63]. Plant defensive responses are triggered in part by compounds regurgitated from the caterpillar gut during feeding [69] or secreted in the saliva [70]. Chewing on petioles would appear to be a particularly effective way to transmit these elicitors throughout the plant. Thus, the ubiquitous wounding of petioles and midribs by tree-feeding caterpillars is surprising.

Caterpillars that clipped leaves or cut furrows and girdles invariably applied saliva to the wounds. In some caterpillar species, salivary constituents are known to suppress defensive pathways, thereby allowing the caterpillars to feed stealthily [69,71]. Almost all research on the impacts of these salivary effectors has been done on herbaceous plants such as tobacco, tomato, maize, and *Arabidopsis* [69,71], but two recent studies indicate that the saliva of tree-feeding caterpillars has similar constituents and similar effects on plant defenses. The first study documented that silkworms (*Bombyx mori*, Bombycidae) apply droplets of saliva to leaf edges during feeding on mulberry (*Morus alba*, Moraceae); the droplets form threads of silk visible in the SEM [27]. An enzyme in the saliva suppresses release of green leaf volatiles, thereby reducing oviposition by a tachinid parasite [27]. Many of the tree genera that are clipped by caterpillars also respond to damage by increasing volatile emissions [72–75]. Thus, salivary constituents entering petiole stubs, furrows, and girdles may serve in part to reduce the release of volatiles attractive to caterpillar enemies. As the saliva dries, silk strands and/or other salivary components might also seal the wounds, thereby directly blocking volatile release.

The second study found that glucose oxidase occurs at high levels in the saliva of both tree and crop-feeding caterpillars [76]. Glucose oxidase catalyzes the conversion of glucose to hydrogen peroxide ($H_2O_2$) and gluconic acid. Hydrogen peroxide acts as a second messenger in plants; it is involved in signaling responses to diverse stresses including wounding [70,77]. In tobacco, glucose oxidase applied to damaged leaves reduced synthesis of nicotine [24,30]

and decreased release of volatiles due to stomatal closure [26]. The leaf-clipping species that have been tested [76] all have high glucose oxidase activity in their salivary glands (Table 1 species #1, 4, 8, 11, 13, 16, 17, 19, 33 in the Papilionidae, Nymphalidae, Notodontidae and Saturniidae). Several additional genera with known clippers in the Sphingidae and Noctuidae also have substantial levels (Table 1 species #24–28, 31). Notably, *Coelodasys unicornis*, which clips leaves and cuts girdles and furrows (S12 and S21 Movies), has high glucose oxidase activity in its saliva (107.0 ± 53.9 nml/min/mg) [76]. If peroxide suppresses direct and indirect defenses in trees, as documented with tobacco, then leaf-clipping caterpillars that apply glucose oxidase to petiole stubs could benefit both by maintaining the acceptability of adjacent leaves where they feed next and by reducing plant emission of volatiles.

The effects of salivary constituents on crops can be highly variable depending on plant species. Glucose oxidase, for example, reduces defenses in tobacco, but increases the production of defensive protease inhibitors in tomato [24,70,78]. Yet tree-feeding caterpillars on diverse trees share the common use of leaf-clipping and saliva application. Io moth caterpillars (*Automeris io*, Saturniidae), for example, clipped leaves and applied saliva to petioles of *Sassafras albidum* in the Laurales (a magnoliid) and to *Prunus serotina* and *Celtis laevigata* in the distantly related Rosales (a fabid) (Table 1) [DE Dussourd Unpublished]. The similar behavior of diverse caterpillar groups on their disparate host plants suggests common defensive pathways in hardwood trees and a shared vulnerability to salivary constituents introduced by leaf clipping. Similar defenses in distantly related trees should facilitate host range expansion by tree folivores. Indeed, many of the leaf-clipping caterpillars in Table 1 (especially species #2, 11–13, 16–20, 25–27, 29, 30, 32–34) feed on trees in more than one family [31,32,37].

Aside from altering plant defensive responses, leaf clipping, furrowing and girdling might also affect nutrient uptake by caterpillars. Girdling, in particular, could prevent the export of photosynthates from distal leaves, thus increasing their nutritional value [12]. Caterpillars generally avoid feeding on veins, which are substantially tougher than leaf lamina [79]. However, in potato, *Solanum tuberosum*, the leaf midrib and side veins contain starch and other carbohydrates [80]. Nutrients sequestered within midribs may explain why some caterpillars ate whole leaves including the midrib (Table 2) instead of clipping the leaves and allowing the midribs to drop to the ground.

## Do salivary constituents enter vascular tissues?

Saliva applied to petiole stubs, furrows, and girdles could reduce defensive responses in nearby leaves if salivary components or their enzymatic products travel in the xylem or phloem. With *Symmerista* leaf clips and furrows, cross sections of petioles and midribs documented that the red salivary pigment entered the xylem (Figs 5A and S2). The red pigment could conceivably have been secreted by larvae during earlier feeding on the leaf. But experiments with safranin O demonstrated that dye applied to artificially clipped petioles and to simulated furrows travelled rapidly in the xylem under both laboratory and field conditions (Figs 5C, S2 and S3). In the lab, dye applied to petiole stubs of sugar maple moved up to 5 cm in 5 minutes. Reducing transpiration by removing all leaves decreased the distance that the dye travelled in isolated petioles demonstrating that diffusion is inadequate to create such rapid movement. As with *Symmerista* pigment, the dye was concentrated in the xylem. Only some of the xylem vessels were stained, as documented in other studies using tracer dyes to analyze xylem transport in tree petioles [81,82].

Retrograde movement of dye in the xylem is opposite the normal flow from stem to leaves. Previous studies on transport of tracer dyes have also documented reversal of flow in response to wounding. In tomato, Lucifer Yellow dye applied to cuts in cotyledons travelled 5mm/sec in

the xylem into portions of leaves with a direct vascular connection to the wounded cotyledons [83]. Dye applied to severed petioles of aspen leaves (*Populus tremula x tremuloides*) moved down the petiole to distal leaves on the branch within 4 hours, but travelled more slowly into leaves closer to the trunk and into leaves on an adjacent branch [84]. Salivary constituents secreted by leaf-clipping caterpillars may similarly move from petiole stubs into adjacent leaves and branches.

With leaf clipping and furrowing, the xylem vessels are directly exposed, which should facilitate entry of salivary compounds and products such as $H_2O_2$. But with girdling, the xylem remains intact, at least with *O. leptinoides* girdles on birch [51]. Without damage to the xylem, safranin O dye did not enter the xylem. However, $H_2O_2$ is a diffusible signal able to pass through cell walls [77,85]. Thus, even girdles may allow entry of salivary products such as $H_2O_2$ into the xylem.

In addition to introducing salivary constituents into the xylem, leaf clipping, furrowing, and girdling undoubtedly cause many other changes in tree physiology that may include initiating hydraulic, electrical, calcium ($Ca^{2+}$), reactive oxygen species (ROS), and hormonal signals [59,86]. Some of these responses such as hydraulic surges occur within seconds, others such as changes in levels of jasmonic acid and other hormones occur within minutes or hours. Identifying how petiole severance and caterpillar secretions affect plant signals has the potential to clarify fundamental defensive responses shared by diverse tree species. *Symmerista albifrons* caterpillars offer a promising new model system, not only because of their colorful saliva and propensity to clip leaves readily in the lab, but also because they undergo periodic outbreaks on oaks. All three species of *Symmerista* in the eastern U.S. have been reported to cause extensive defoliation [33,87–89].

## Conclusion

This study documents that diverse caterpillars clip leaves and bathe the petiole stubs in saliva; some species clip midribs and petioles repeatedly. These results do not exclude the possibility that leaf-clipping caterpillars benefit from reduced predation by eliminating visual cues associated with their feeding. However, the visual cue hypothesis alone is inadequate to explain caterpillar behavior. Insights from future research that identifies exactly how petiole clips and saliva alter tree physiology will likely apply to thousands of insect-tree interactions that may involve sawflies as well as caterpillars.

## Supporting information

**S1 Fig. Feeding damage of caterpillars on trees and vines.** (A) Clipped leaves of sycamore (*Platanus occidentalis*, Platanaceae) and willow oak (*Quercus phellos*, Fagaceae) and clipped leaflet of pecan (*Carya illinoinensis* leaf, Juglandaceae) found on the ground in Conway, Arkansas. (B) Larva of *Lochmaeus manteo* (Notodontidae) dispersing from a willow oak tree (*Q. phellos*) to pupate. The ground is covered with clipped oak midribs and caterpillar frass due to an *L. manteo* outbreak June 2013 in Conway, Arkansas. (C) Two larvae of *Darapsa myron* (Sphingidae) on excised stems of the vine *Ampelopsis cordata* (Vitaceae). The larvae ate entire leaf blades, then sequentially consumed portions of the petiole and rubbed saliva on the petiole stubs after finishing each segment. The larvae never clipped the petioles despite the similar appearance of the *A. cordata* petiole stubs to petioles clipped by other caterpillar species (Figs 1–3 and S4). (D) Final instar of *Symmerista leucitys* feeding on sugar maple, *Acer saccharum* (Sapindaceae). The larva did not clip the petiole, but instead applied red saliva to each major leaf vein after consuming a portion of the leaf.
(TIF)

**S2 Fig. Cross sections of white oak petioles (*Quercus alba*, Fabaceae) with xylem stained by *Symmerista albifrons* pigment or safranin O dye.** (A) Petiole that was clipped by a final instar of *S. albifrons*, then severed with a razor blade five minutes later 2.5mm from the petiole tip. The red salivary pigment moved into the petiole staining the xylem. (B) Petiole xylem stained by safranin O dye (0.2%) pulled up the xylem by transpiration. (C) Petiole stub treated with 1µl safranin O dye (0.2%). After 5 minutes, the petiole was severed 2.5mm from the initial cut. The dye moved in a retrograde direction down the xylem staining the same tissues as *S. albifrons* pigment in (A) and dye in (B). The petiole slices in A-C were photographed dry to prevent pigment and dye from dissolving or spreading to adjacent tissues; some distortion of the exterior surface of the slices resulted. (D) Thin slice of petiole stained with astra blue followed by safranin O to illustrate the position of vascular tissues. The phloem is located between the xylem and fiber band (sclerenchyma) (AL Filartiga pers. comm.). Scale bars equal 0.3 mm.
(TIF)

**S3 Fig. Retrograde movement of dye down a petiole on a sugar maple tree (*Acer saccharum*, Sapindaceae).** The petiole was severed, then 1µl safranin O dye (0.2%) was placed on the petiole stub. After 5 minutes, cross sections of the petiole were cut 2.5, 5, 10 and 20 mm down the petiole. The dye moved in some xylem vessels over 20 mm in five minutes. Scale bars equal 0.3 mm.
(TIF)

**S4 Fig. Cluster of *Datana integerrima* larvae feeding on a pecan tree (*Carya illinoinensis*, Juglandaceae).** The larvae consumed entire leaflets except for the midribs, then repeatedly clipped each midrib leaving only the rachis and short stubs.
(TIF)

**S1 Data. Time caterpillars spent clipping leaves and applying saliva.**
(XLSX)

**S2 Data. Time non-clipping caterpillars spent applying saliva.**
(XLSX)

**S3 Data. Distance dye and pigment moved in the xylem plus leaf measurements.**
(XLSX)

**S1 Movie. Final instar *Symmerista albifrons* (Notodontidae) clipping the petiole of white oak (*Quercus alba*, Fagaceae) at 2x speed.** The larva applied red saliva to the basal side of the petiole, not to the portion that falls to the ground.
(MP4)

**S2 Movie. Final instar *Symmerista albifrons* applying saliva in slow motion (0.5x).** The larva clipped the petiole of a white oak leaf (*Quercus alba*, Fagaceae). Red saliva can be seen stretching three times from the spinneret to the petiole surface.
(MP4)

**S3 Movie. Final instar *Symmerista albifrons* with cauterized spinneret clipping a white oak petiole (*Quercus alba*) at 2x speed.** Cautery blackens the spinneret and prevents release of red saliva from the labial salivary glands. The larva still wiped its spinneret on the petiole stub, but no fluid appeared. The same larva before cautery applied copious red saliva (S2 Movie).
(MP4)

**S4 Movie.** *Symmerista leucitys* **larva applying red saliva to a furrow.** After cutting the furrow in a sugar maple leaf (*Acer saccharum*), the final instar larva spent over 70 seconds secreting saliva onto the furrow (the last 35 seconds are shown).
(MP4)

**S5 Movie.** *Symmerista albifrons* **applying saliva during feeding on willow oak (***Quercus phellos***).** A red labial salivary gland is visible through the transparent body wall between the second and third pair of legs.
(MP4)

**S6 Movie.** *Symmerista albifrons* **unsuccessfully clipping a white oak midrib (***Quercus alba***).** The final instar larva was unable to penetrate the slippery, round midrib so it moved closer to the tip of the midrib and successfully clipped it. The larva then returned to the original site and again failed to cut into the midrib. The caterpillar then clipped the midrib a second time closer to the tip (stub shown in video), then returned just proximal to its previous unsuccessful attempt and again was unable to sever the midrib as shown in the video.
(MP4)

**S7 Movie. Saliva of four caterpillar species stretching from the opening of the labial salivary gland to the plant surface.** Clear saliva is visible in all four caterpillar species: *Furcula borealis* final instar after clipping a *Prunus serotina* petiole, *Heterocampa umbrata* final instar after clipping a *Quercus nigra* petiole, *Ceratomia undulosa* final instar after clipping a *Fraxinus pennsylvanica* midrib, and *Automeris io* penultimate instar that did not clip a *Celtis laevigata* petiole.
(MP4)

**S8 Movie. Red-spotted purple larva (***Limenitis arthemis***, Nymphalidae) clipping the petiole of a ***Prunus serotina*** leaf (Rosaceae) at 2x speed.** The final instar larva spent 200 seconds applying saliva to the petiole stub; 53 seconds are shown.
(MP4)

**S9 Movie. Final instar** *Paraeschra georgica* **(Notodontidae: Notodontinae) clipping the petiole of a** *Quercus nigra* **leaf (Fagaceae) at 2x speed.**
(MP4)

**S10 Movie.** *Furcula borealis* **(Notodontidae: Cerurinae) clipping the petiole of a** *Prunus serotina* **leaf (Rosaceae) at 2x speed.** The final instar larva spent 99 seconds applying saliva to the petiole stub; 41 seconds are shown.
(MP4)

**S11 Movie.** *Datana angusii* **(Notodontidae: Phalerinae) clipping a leaflet midrib of** *Carya illinoinensis* **(Juglandaceae).** The final instar larva started to depart twice, but returned each time to apply additional saliva to the midrib stub. Film speed 2.5x.
(MP4)

**S12 Movie. Final instar** *Coelodasys unicornis* **(Notodontidae: Heterocampinae) clipping the petiole of a** *Betula nigra* **leaf (Betulaceae) shown at 4x speed.**
(MP4)

**S13 Movie. Final instar** *Acronicta impleta* **(Noctuidae) clipping the petiole of a** *Carya illinoinensis* **leaflet (Juglandaceae) at 2x speed.** The larva spent 427 seconds cutting the petiole and 220 seconds applying saliva to the petiole stub; 26 seconds of petiole clipping and 49

seconds of saliva application are shown.
(MP4)

**S14 Movie. *Ceratomia amyntor* (Sphingidae: Sphinginae) clipping *Ulmus americana* (Ulmaceae).** The final instar spent 115 seconds cutting the petiole and 155 seconds applying saliva to the petiole stub; 13 seconds of petiole clipping and 21 seconds of saliva application are shown.
(MP4)

**S15 Movie. *Anisota virginiensis* (Saturniidae: Ceratocampinae) applying saliva to a petiole stub of *Quercus shumardii* (Fagaceae).** The final instar larva did not cut completely through the petiole; the distal portion can be seen dangling from the petiole stub.
(MP4)

**S16 Movie. Penultimate instar *Automeris io* (Saturniidae: Hemileucinae) clipping a *Celtis laevigata* stem (Cannabaceae) two times (shown at 3x speed).** The larva applied saliva for 49 seconds after the first stem clip and for 83 seconds after the second.
(MP4)

**S17 Movie. Penultimate instar of *Actias luna* (Saturniidae: Saturniinae: Saturniini) clipping a pecan leaflet (*Carya illinoinensis*, Juglandaceae) shown at 2.5x speed.**
(MP4)

**S18 Movie. *Hyalophora cecropia* (Saturniidae: Saturniinae: Attacini) clipping the petiole of a *Prunus serotina* leaf (Rosaceae) shown at 3x speed.** The penultimate instar larva spent 55 seconds cutting the petiole and 108 seconds applying saliva to the petiole stub; 25 seconds of petiole clipping and 85 seconds of saliva application are shown.
(MP4)

**S19 Movie. Final instar *Darapsa myron* (Sphingidae: Macroglossinae) applying saliva to petiole stub of *Ampelopsis cordata* (Vitaceae) without clipping the petiole.**
(MP4)

**S20 Movie. *Heterocampa umbrata* chewing a furrow in a *Quercus falcata* midrib, then applying saliva to the furrow.** Afterwards the final instar larva resumed feeding beyond the furrow. Video shown at 2x speed.
(MP4)

**S21 Movie. Final instar of *Coelodasys unicornis* (Notodontidae: Heterocampinae) girdling a *Betula nigra* stem (Betulaceae), then applying saliva to the girdle.** Video shown at 5x speed.
(MP4)

**S22 Movie. *Schizura ipomoeae* (Notodontidae: Heterocampinae) crimping an *Acer rubrum* petiole (Sapindaceae) in a ring and applying saliva.** After secreting saliva with a back and forth head motion, the larva left the petiole and initiated feeding on another leaf that it had previously crimped.
(MP4)

## Acknowledgments

Special thanks to Samuel Jaffe, Tucker Cooley, and Charlotte Cooley for providing *Symmerista* eggs and larvae and to Jim Miller for identifying *S. albifrons* from West Virginia. Thanks also to all who allowed me to place mercury vapor and UV lights on their property to collect female

moths (Reid and Ginny Adams, Ben Cash, Steve Karafit, K.C. Larson, Jerry Mimms, Robert Parker, Ed Romero and Nicole Hardiman, Steve and Nelle Runge, Buffalo National River at Steel Creek, Camp Robinson WMA, Ouachita Mountain Biological Station, Pine Hollow Natural Area, Scott Henderson Gulf Mountain WMA, and Woolly Hollow State Park). I am also grateful to David Wagner and Erin Wiley for providing helpful comments on the manuscript, to Rick Noyes for assistance identifying plants, to David Wagner, Steven Passoa, Bri Trejo, and Madison Srebalus for help acquiring moths, and to Madison Srebalus for assistance rearing caterpillars. Permits were kindly provided by USDA APHIS, Arkansas Game and Fish Commission, DOI National Park Service, Arkansas Natural Heritage Commission, and the Arkansas Department of Parks and Tourism.

## Author Contributions

**Conceptualization:** David E. Dussourd.

**Funding acquisition:** David E. Dussourd.

**Investigation:** David E. Dussourd.

**Methodology:** David E. Dussourd.

**Visualization:** David E. Dussourd.

**Writing – original draft:** David E. Dussourd.

**Writing – review & editing:** David E. Dussourd.

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
