## [Decision Letter · Decision Letter 0]

3 Mar 2022

Salivary surprise: Symmerista caterpillars anoint petioles with red saliva after clipping leaves

PONE-D-21-41020

Dear Dr. Dussourd,

We’re pleased to inform you that your manuscript has been judged scientifically suitable for publication and will be formally accepted for publication once it meets all outstanding technical requirements. 

Kind regards,

Livia Maria Silva Ataide

Academic Editor

PLOS ONE

Journal REquirements:

1. We note that you have referenced (Dussourd unpub. data) which has currently not yet been accepted for publication. Please remove this from your References and amend this to state in the body of your manuscript: (ie “Dussourd et al. [Unpublished]”) as detailed online in our guide for authors

Additional Editor Comments (optional):

Dear authors, congratulations for the nice manuscript. Please address the comments listed by Reviewer #1

Reviewers' comments:

Reviewer's Responses to Questions

**Comments to the Author**

1. Is the manuscript technically sound, and do the data support the conclusions?

Reviewer #1: Yes

Reviewer #2: Yes

2. Has the statistical analysis been performed appropriately and rigorously? 

Reviewer #1: Yes

Reviewer #2: Yes

3. Have the authors made all data underlying the findings in their manuscript fully available?

Reviewer #1: Yes

Reviewer #2: Yes

4. Is the manuscript presented in an intelligible fashion and written in standard English?

Reviewer #1: Yes

Reviewer #2: Yes

5. Review Comments to the Author

Reviewer #1: The manuscript is a well written scientific document that provides valuable information in the field of entomology. All analyses were appropriate for the observed data, and the data used in the study were provided.

Reviewer #2: The paper is masterfully written and an impressive comparison among multiple caterpillars looking at leaf clipping behavior in caterpillars. The author provides strong evidence that leaf clipping behavior in caterpillars on the leave's petiole is for the delivery of saliva that likely reduces the induction of plant defenses. Challenging the previous suggestion that leaf clipping is to avoid predator's such as birds. The paper is truly impressive in scope, really well organized and comprehensive. Going to a broad behavior phenomenon to demonstrating caterpillar salivary movement in the plant. I think this paper has high importance and should be published with no obvious revision of any sort.

6. PLOS authors have the option to publish the peer review history of their article (what does this mean?). If published, this will include your full peer review and any attached files.

Reviewer #1: **Yes: **A. Daniel Greene

Reviewer #2: **Yes: **Richard O. Musser

---

## [Editor Report · Acceptance letter]

8 Mar 2022

PONE-D-21-41020 

Salivary surprise: *Symmerista* caterpillars anoint petioles with red saliva after clipping leaves 

Dear Dr. Dussourd:

I'm pleased to inform you that your manuscript has been deemed suitable for publication in PLOS ONE. Congratulations! Your manuscript is now with our production department. 

Kind regards, 

on behalf of

Dr. Livia Maria Silva Ataide 

Academic Editor

PLOS ONE